# OpenReview forum: "Constitutional Black-Box Monitoring for Scheming in LLM Agents"
_ICML.cc/2026/Conference — ICML 2026 regular_

### Official Review · Reviewer_Ktw7 · 2026-03-10

**Soundness:** 2
**Presentation:** 2
**Significance:** 2
**Originality:** 2
**Overall Recommendation:** 3
**Confidence:** 3

**Summary:**

This paper proposes constitutional black-box monitors: prompted LLM classifiers that detect agent scheming using only externally observable inputs and outputs. The authors introduce two synthetic data generation pipelines (STRIDE and Gloom) to train these monitors without real-world scheming examples, and evaluate on semi-realistic ControlArena trajectories. Key findings are that synthetic data contains a transferable scheming signal, but performance saturates quickly with simple prompt sweeps matching more extensive optimisation.

**Compliance With Llm Reviewing Policy:**

Affirmed.

**Final Justification:**

The rebuttal and authors' follow-up comment have been noted. The distributional gap analysis, ControlArena ceiling result, and fine-tuning baseline are useful additions that strengthen the empirical story. Nevertheless, my novelty concern remains. STRIDE and Gloom, while carefully engineered, function primarily as data utilities rather than advances in detection methodology: they generate labeled trajectories for prompt optimization but do not themselves constitute a new mechanism for detecting scheming. The key empirical finding that prompt sweeping suffices and heavier optimization overfits is honest and practically useful, but does not in itself clear the bar for a novel technical contribution: the result characterizes the limits of an existing optimization paradigm rather than introducing a new one.

**Key Questions For Authors:**

- How exactly does the synthetic data interact with the realistic ControlArena sandbox environments? The bridging between synthetic generation and semi-realistic evaluation is not clearly described in the main paper.

- The paper claims in conclusion to introduce pipelines "for training constitutional black-box monitors," but no gradient-based training of the monitor LLM appears to take place, the monitor is optimised purely through prompt selection. Could the authors clarify what "training" means in this context?

**Limitations:**

Yes

**Strengths And Weaknesses:**

Strengths:

- Well-structured introduction, motivating the problem well from broad to narrow.
- STRIDE and Gloom offer the community two complementary approaches for generating synthetic scheming datasets.
- The practical conclusion, that simple prompt sweeps suffice and refinement risks overfitting, is a valuable and honest takeaway for practitioners.

Weaknesses:

- Limited novelty: the core contribution reduces to prompt sweeping over existing frontier models, which is a relatively low bar for innovation.
- The central claim that "constitutional black-box monitors can detect scheming without access to real-world examples" is weakened by the lack of systematic analysis of how dissimilar the synthetic data actually is from the real-world trajectories. If the two distributions are already close, the claim loses power.
- The saturation finding may not generalise, as the authors themselves acknowledge: it may reflect that the synthetic distribution is too narrow or artifact-heavy than a fundamental limit of the prompting approach. The paper provides no evidence to the contrary.
- Pipeline details are largely put in the appendix; the main paper would benefit from at least one illustrative example of how scheming trajectories are constructed (more detailed than Figure 2).
- Results also appear too late (page 7 only), with the experimental setup taking disproportionate space.

---

> ### Author Rebuttal · Authors · 2026-03-31
>
> Thank you for your thoughtful and detailed review. We appreciate your recognition of STRIDE and Gloom as valuable synthetic data pipelines, and your acknowledgement of the practical insights of our findings. We also thank you for the constructive questions and suggestions, which we address below and will incorporate into the revised PDF.
>
> **Core contribution.** While prompt sweeping is the optimization technique we ultimately recommend to practitioners, we see our main contribution as an end-to-end framework for monitoring: novel pipelines for scalably generating synthetic training data (STRIDE and Gloom), a suite of optimization approaches (along with recommendations for practitioners), and a semi-realistic evaluation protocol with strict train-test separation, partial AUROC evaluation, and high-confidence execution-based labels. We see the combination of these contributions as a meaningful step forward in LLM agent monitoring.
>
> **The synthetic vs. semi-realistic gap.** We believe our results support the claim that “constitutional black-box monitors can detect scheming without real-world examples,” regardless of whether there is a distributional shift between our synthetic data and test set. If there is no shift, the claim holds because the synthetic data is sufficiently representative to catch the scheming in our test data, which represents the most realistic form of scheming currently available for evaluation. However, we agree that our result is more compelling if it also demonstrates robustness to distribution shift, which requires analysis of the differences between the training and test data. To address this, we compared train and test distributions, including qualitative task/environment breakdowns and statistics such as trajectory length and number of tool calls. We refer you to our response to Reviewer 2gi4 ("Comparison of datasets") for a detailed summary of these differences, which span trajectory length, token density, tool usage patterns, and the qualitative structure of scheming behavior. We believe these show clear distributional differences between the synthetic training data and the ControlArena test set.
>
> **Generality of the saturation finding.** We fully agree that the observed saturation may reflect limitations in the generality of our synthetic datasets rather than (or as well as) limitations of our prompting approach. We note that we observe a similar performance ceiling across two structurally different pipelines (STRIDE and Gloom), which suggests the effect is not tied to a single data source. We agree that evidence to better diagnose the cause of the performance saturation and thus understand how well it might generalise would be valuable, and we refer you to our rebuttals to reviewer 2gi4 (“Diagnosing the performance ceiling.” section) in which we outline two experiments we have conducted to better diagnose the performance ceiling. We include these results in the revised PDF.
>
> **Pipeline clarity and presentation.** We appreciate the suggestion to improve pipeline clarity in the main text. In the revised version, we include a concise, illustrative example in the main text that walks through a single trajectory (from constitution → generation process → labeled trajectory), complementing Figure 2. To preserve space, this example is intentionally lightweight and focused on intuition, with full pipeline details and additional examples remaining in the appendix. We also acknowledge the concern that the results appear only on page 7. In the revised PDF, with the addition of further experimental results (referred to in the previous paragraph), we rebalance the paper so that the results section occupies a larger portion of the main 8 pages.
>
> **Clarifying the relationship between synthetic data and ControlArena.** All data sources (STRIDE, Gloom, and ControlArena) produce transcripts in a common format (sequences of user messages, tool calls, and tool outputs) with positive or negative labels for scheming. We treat the synthetic trajectories as labelled examples for monitor optimisation, and then evaluate the resulting monitors on the ControlArena test set. By design, there is no direct interaction between synthetic data and the ControlArena environments: ControlArena is used solely to generate held-out evaluation trajectories. The implicit link between the two is the constitution itself, which we use as both the generative specification for synthetic data and the reference against which we validated the ControlArena test set (e.g., ensuring there are no contradictions in our labels).
>
> **“Training” terminology.** Thank you for pointing this out. We agree that the use of the word “training” is confusing here since our main methodologies are prompt sweeps, refinement, and DSPy. We have replaced the word with “optimizing” in the revised PDF.
>
> We appreciate your feedback and believe these clarifications will strengthen the paper. We would be happy to address any further questions.

---

> > ### Author Rebuttal · Reviewer_Ktw7 · 2026-04-03
> >
> > I appreciate the authors' clarifications on the synthetic-to-real distributional gap and the terminology correction. The scaling analysis and in-distribution ControlArena ceiling result (referenced in response to Reviewer 2gi4) are informative and do strengthen the saturation finding: the result that optimizing on in-distribution ControlArena data gives comparable performance to the synthetic ceiling is a meaningful diagnostic.
> >
> > However, my core concern about limited novelty remains unaddressed. The central methodological contribution reduces to prompt sweeping over existing frontier models, with STRIDE and Gloom serving primarily as data generation utilities rather than novel monitoring architectures. The paper's most actionable finding, that simple grid search suffices and heavier optimization overfits, is a useful empirical observation, but does not in itself constitute a sufficiently novel technical contribution for acceptance. The rebuttal clarifies and supplements but does not materially change this assessment. I maintain my score.

---

> > > ### Author Response · Authors · 2026-04-07
> > >
> > > Thank you for the clarification - we appreciate you outlining your perspective on novelty. We respectfully disagree that our central methodological contribution reduces to prompt sweeping over existing frontier models.
> > >
> > > In our view, the primary challenge in building monitors for LLM scheming is the lack of suitable data and evaluation protocols. STRIDE and Gloom provide novel, constitution-driven frameworks for synthetic data generation, each of which constitutes a meaningful technical contribution in its own right. Our selection of ControlArena settings aligned with our scheming definition, together with our execution-based labeling procedure and use of the partial AUROC metric, provides a high-confidence evaluation suite. This end-to-end setup is a core part of our methodological contribution.
> > >
> > > While prompt sweeping is the optimization approach we ultimately recommend, it is not the only approach we study. We evaluate human-guided refinement, DSPy-based optimization (GEPA and MIPROv2), and (in the revised PDF) a fine-tuning baseline. The finding that simple prompt sweeps match or outperform more complex optimization methods in this setting is non-obvious, and we view this comparison - not the prompt sweep method itself - as the key empirical contribution.

---

### Official Review · Reviewer_wCe6 · 2026-03-12

**Soundness:** 3
**Presentation:** 4
**Significance:** 3
**Originality:** 3
**Overall Recommendation:** 5
**Confidence:** 3

**Summary:**

This paper studies constitutional black-box monitors for detecting scheming. A constitutional black-box monitor is a frontier LLM prompted to flag certain behavior in a transcript. Scheming is when a model attempts to carry out a secret goal while not being detected. The paper generates 1000 synthetic examples of scheming from a definition (constitution) of scheming, and then uses prompt optimization to find the best black-box monitor. The authors find good transfer from synthetic to more realistic environments, but that the performance of the detectors saturates far from 100% detection success.

**Compliance With Llm Reviewing Policy:**

Affirmed.

**Final Justification:**

As described in my reply to the authors, I think this is a good piece of work and should be accepted. Some limitations remain, but the work overall is above the bar.

**Key Questions For Authors:**

+ Have you tried fine-tuning a smaller model to classify the transcripts? It would be interesting to see the performance of either an open-weight model or a closed-source model with finetuning API.
+ Do you have an idea of what key properties/tells the monitors are generally using in order to distinguish scheming from non-scheming?

**Limitations:**

Yes

**Strengths And Weaknesses:**

# Strengths
+ The problem is important and well-motivated. The threat model is clear, and although the affordances are very restricted (only prompt-based access to frontier models), the overall setting is internally consistent.
+ The methodology of splitting synthetic and more realistic test data is a rigorous and consistent with the threat model.
+ The main results are presented with limitations clearly addressed. Although the mainline results with regards to prompt optimization are negative results, they are clearly presented.
+ Testing both the STRIDE and GLOOM methods is a comprehensive way to marginalize over specific differences in how a synthetic dataset might be constructed. It is very useful from a practitioner's point of view to know that there is, at a high level, not much difference between STRIDE and GLOOM.

# Weaknesses

+ There are no comparison to fine-tuned monitor baselines. The paper evaluates only prompted monitors and identifies a performance ceiling at ~0.85 partial AUROC, but doesn't test whether fine-tuning a (potentially much smaller) model on the 2,000 synthetic examples would break through this ceiling. This should not be a very expensive experiment to perform, since fine-tuning APIs are readily available for open-weight models and even closed-weight models like some GPT models. This would straightforwardly distinguish between the limitation being a problem of prompting and the limitation being the synthetic data itself. The authors argue this can be future work but I think the paper is somewhat lacking for not including it when it is relatively easy and inexpensive to do, and has a direct implication on the paper's takeaways.
+ Limited analysis of what the monitors actually detect. The paper doesn't provide much insight into what features of the trajectories drive the monitor's classification. Are monitors detecting specific tool patterns, unusual sequences of actions, or a general 'vibe'?  A more systematic analysis, such as by classifying the false positives/negatives via an LLM, might give some useful insights for future work.

---

> ### Author Rebuttal · Authors · 2026-03-31
>
> Thank you for your thoughtful and constructive review. We appreciate your positive assessment of the problem framing, experimental design, and clarity of presentation, as well as your recognition of the value of comparing STRIDE and Gloom. We also thank you for the insightful suggestions, which we address below and incorporate into the revised PDF.
>
> **Fine-tuning baseline.** We agree this is a valuable comparison. We fine-tuned Qwen3-8B on STRIDE and Gloom, achieving 0.549 and 0.555 pAUROC respectively, substantially above the prompted Qwen3-8B baseline (0.277) but below prompted frontier monitors. We describe the full setup and planned extensions for the camera-ready paper (larger models, hyperparameter tuning, like-for-like prompting vs. fine-tuning comparison) in our response to Reviewer bxU3.
>
> **Understanding what monitors detect.** We agree that better qualitative understanding of the features driving monitor decisions is valuable. In light of this, we conducted qualitative analysis of common features appearing in our monitors’ reasoning traces. This analysis highlighted a range of features that monitors commonly pick up on in trajectories, including: (i) unjustified code complexity (e.g. unnecessary helper functions), (ii) inconsistencies between stated behaviour in the user message and actual tool use behaviour, (iii) special-casing in code (suggestive of backdoors), and (iv) blabbing in code comments (e.g. comments mentioning that the code is intended to be “subtle”). We have also conducted some analysis of why monitors fail: weaknesses include "weaponized transparency" (agent disclosing actions while framing them benignly), plausible-sounding technical justifications for actions that serve hidden goals, and missing scheming actions disguised as “helpful overreach”. We have included a detailed breakdown of this qualitative analysis in the appendix of the revised PDF.
>
> We appreciate your feedback and believe these additions will strengthen the paper. We would be happy to address any further questions.

---

> > ### Author Rebuttal · Reviewer_wCe6 · 2026-04-02
> >
> > Thank you for the response, especially with the fine-tuning details. I will maintain my score at 'accept'.

---

### Official Review · Reviewer_2gi4 · 2026-03-16

**Soundness:** 3
**Presentation:** 3
**Significance:** 3
**Originality:** 2
**Overall Recommendation:** 5
**Confidence:** 4

**Summary:**

This paper addresses the challenge of detecting scheming in LLM agents—where agents covertly pursue misaligned goals while appearing cooperative. The authors develop constitutional black-box monitors: prompted LLM classifiers that analyze only externally observable behavior (tool calls and outputs), excluding internal chain-of-thought. They introduce two synthetic data generation pipelines (STRIDE and Gloom) to create training and validation trajectories for follow up optimization studies from natural language scheming specifications, then evaluate on 7,500 held-out real-world trajectories from ControlArena where ground truth is determined by executable environment checks. Key findings show that monitors trained on synthetic data can generalize to more realistic settings, but performance saturates quickly—simple prompt sweeps match heavily-optimized approaches, and aggressive optimization leads to overfitting.

**Compliance With Llm Reviewing Policy:**

Affirmed.

**Final Justification:**

Increasing score to 5 as rebuttal addressed main concerns around data size impacting performance ceiling and difference and contributions of using two different synthetic pipelines for generating training data.

**Key Questions For Authors:**

1. What is the complementarity of STRIDE vs Gloom as approaches for synthetic data generation? The paper introduces two approaches for synthetic data generation but does not go into depth to compare and contrast them beyond Figure 4. As this is one of the core contributions of identifying the use of generating synthetic datasets for real-eval sets, what is the reasoning behind using either or both approaches and do they fill different coverage gaps?

2. The paper identifies a performance ceiling for optimization techniques, but beyond varying the optimization techniques what analysis has been conducted to further diagnose the performance ceiling? For instance, is this the result of a limited train dataset size, is this the result of the synthetic train set being out-of-distribution to the eval set or having critical coverage gaps (are we misclassifying consistently the same attack vectors, etc.)?

3. It is mentioned that in the test set one discarded unsuccessful scheming attempts, could potentially these cases lead to more borderline examples of scheming that would be provide more discriminatory signals across black-box monitoring models? Could you report the performance on the discarded trajectories, or clarify what fraction of side-task assignments failed and why?

**Limitations:**

yes

**Strengths And Weaknesses:**

Overall, the paper presented a solid methodology to evaluate the transfer learning of synthetic training/validation of black-box monitors to realistic test sets

Soundness:

*Strengths*
- Rigorous split of train/test separation, with focus on curating a test set based on real-world agent traces and leveraging executable, deterministic scheming tasks to avoid additional bias from LLM-based judges
- Metrics leveraged sound reasonable given monitoring context. Leveraging partial AUROC for FPR < 0.2 and use of confidence intervals to account of non-deterministic nature of LLM-based monitoring agents
- Experiments with multiple optimization approaches, scaling in complexity/automation, with additional, complimentary experiments included in the Appendix

*Weaknesses*
- Limited synthetic data scale: the paper focuses on a fixed synthetic dataset of 1000 per methodology (relative to the 7,500 eval set) and notes optimizations leading to overfitting. Potentially the dataset limit is one axis to explore for evaluating optimization ceilings

Presentation:

- *Strengths*: Paper was well written, includes clear figures, and well structured flow

Significance:

*Strengths*
- The paper presents in finding in such a way to create actionable practitioner guidance. The paper compares and contrasts multiple prompt optimization techniques of increasing complexity to identify potential uplifts from synthetic dataset optimizations on real-world eval sets. The findings show lower utility for the more autonomously complex optimization methods, and indicate a favour to more straight forward pipeline with classic grid search techniques
- The paper validates the use of synthetic datasets for train/val and their transfer to real-world eval environments. As particular cases (especially in the safety domain) prove difficult to source from real-world cases, the ability to leverage synthetic sets for training/val and their ability to transfer to real eval sets provides high utility for the general practitioner

*Weaknesses*
- The paper presents two methods for generating synthetic data without a clear recommendation of preference or what gap either method fills relative to the overall pipeline. More exploration into the differences (both quantitatively and qualitatively) in data generated by leveraging either method (diversity, realism, coverage, etc.) and their performance implication on the final eval set can offer stronger direction to practitioners on which method to prioritize in their own synthetic dataset creations

Originality:

- *Strengths*: The work aims to deepen understanding of leveraging and optimizing black-box monitoring systems on synthetic train/val sets and their performance transfer on real eval sets. The work introduces one new methodology for generating synthetic sets (STRIDE) and builds upon an existing methodology Gloom. The work also evaluates multiple prompt optimization techniques to deepen one's understanding of the benefits of leveraging more complex methodologies over standard grid search approaches.
- *Weaknesses*: While the work introduces two new methods to generate synthetic data (the latter Gloom, building upon existing Bloom infrastructure), more exploration to compare and contrast both methods is suggested to further emphasize the novelty of the approaches and their contributions relative to past works. The conclusion of an empirical performance ceiling can also be further investigated by exploring how dataset size, and synthetic dataset coverage influences the ceiling of optimization techniques

---

> ### Author Rebuttal · Authors · 2026-03-31
>
> Thank you for the thoughtful review and helpful suggestions.
>
> **Number of training examples.** We appreciate your comment that the observed performance ceiling may be driven by limited training data. To clarify our choice of training set size, we conducted a scaling analysis prior to fixing N=1,000: we generated larger synthetic datasets using STRIDE and Gloom, and evaluated a representative set of (model, prompt) combinations on subsets of increasing size N, computing partial AUROC and bootstrap confidence intervals. We found that around N=1,000, confidence intervals between candidate monitors became largely non-overlapping and relative rankings stabilized, with larger N yielding only marginal changes in both absolute performance and monitor ordering. This analysis excluded test evaluation to preserve a fully held-out test set, but suggests that larger training sets would yield similar optimized monitors and thus similar test performance. We will include additional details on this analysis in the revised PDF.
>
> **Diagnosing the performance ceiling.** We believe our dataset size analysis, showing only marginal improvements beyond N=1,000, suggests that increasing training set size within the same distribution is unlikely to substantially shift the observed performance ceiling. To assess how much the performance ceiling is attributable to the distribution shift, we optimized a monitor on 1,000 in-distribution (but out-of-sample) ControlArena trajectories using grid search over prompts and three frontier LLMs, followed by one round of human refinement. This yielded 0.81 pAUROC, with 95% confidence interval [0.79, 0.83]. This is comparable to, rather than substantially exceeding, the synthetic-data ceiling, suggesting that the performance plateau is not primarily driven by distribution shift between train and test. We note this result is preliminary: optimization was conducted on a subset of ControlArena environments and limited to one round of human refinement. For the camera-ready version, we will extend to the full optimization pipeline (grid search → human refinement → DSPy) across all environments.
>
> **Comparison of datasets.** We perform a detailed analysis of STRIDE, Gloom, and ControlArena (test set) trajectories, with quantitative and qualitative comparisons included in the revised paper. In summary: STRIDE trajectories are longer-horizon (median 12 turns, \~4.5k tokens), Gloom trajectories are shorter (median 10 turns) but much denser (\~21.8k tokens), and ControlArena is heterogeneous (median 12 turns, range 2–160). Tool usage also differs: STRIDE averages \~1.0 tool call per assistant message, while Gloom averages ~2.6 calls and ControlArena averages \~1.1. Qualitatively, STRIDE trajectories exhibit more localized scheming concentrated in specific tool calls or short action sequences, while Gloom involves more distributed, multi-step scheming that unfolds across the trajectory. STRIDE tasks are more procedural, whereas Gloom tasks are more open-ended, leading to more varied agent behavior in both benign and scheming cases. ControlArena shares properties with both (e.g., trajectories with many simple steps in the bash and IaC settings, long-horizon behavior in SHADE arena) but also includes very short trajectories (e.g., the rogue eval setting).
>
> The performance of the two training pipelines is comparable, with similar gains over our minimal baseline; however, STRIDE shows a slight advantage in prompt sweep transfer (Figure 4) and marginally better performance under human refinement and DSPy optimization (Figure 5). The two pipelines are complementary: observing synthetic-to-real transfer and a performance ceiling across independently developed methods strengthen both findings. We will clarify these comparative points in the revised text.
>
> **Discarded unsuccessful scheming attempts.** We previously explored including unsuccessful scheming trajectories, but found this introduced label ambiguity: in many cases, although the agent was prompted with a side task, there was no observable evidence that the agent attempted to execute it. We therefore made the decision to restrict positive labels to trajectories with verified side-task completion. We collected sufficiently many samples such that we were left with 3,750 positive labels after this discarding process (ensuring a balanced test set). That said, we agree these failed trajectories could provide useful borderline cases if restricted to those for which we can verify evidence of attempted side-task execution. We report the fraction of discarded trajectories (49%) and clarify this decision in the revised paper.
>
> We thank you again for your constructive feedback and believe these additions and clarifications will significantly strengthen the paper. We would be happy to address any further questions.

---

> > ### Author Rebuttal · Reviewer_2gi4 · 2026-04-03
> >
> > Thank you for the follow up insights and revisions to the paper, I will increase my score accordingly

---

### Official Review · Reviewer_iDMb · 2026-03-17

**Soundness:** 3
**Presentation:** 2
**Significance:** 2
**Originality:** 2
**Overall Recommendation:** 4
**Confidence:** 5

**Summary:**

The paper studies black-box monitoring for scheming in LLM agents. It proposes a constitutional monitoring setup where prompted LLM classifiers detect scheming from only externally visible trajectories, without access to chain-of-thought. The main technical contribution is two synthetic data pipelines, Stride and Gloom, used to generate 2,000 synthetic training trajectories. The main finding is that monitors selected on synthetic data do transfer to the more realistic test environments, but performance saturates early: simple prompt sweeps already reach strong performance, and heavier human or Dspy-based optimization often overfits rather than improving generalization.

**Compliance With Llm Reviewing Policy:**

Affirmed.

**Final Justification:**

Main concerns were addressed during the rebuttal.

**Key Questions For Authors:**

1. How much of the observed ceiling is due to synthetic data artifacts versus the limits of prompting itself (maybe a dedicated ablation focused on it would address it)?
2. DSPy optimization uses classification loss, while the main evaluation metric is AUROC based. How much do the conclusions about “overfitting under heavy optimization” depend on this optimization-metric mismatch?
3. Since the monitor excludes CoT by design, are there important failure modes where tool-use traces alone are insufficient, and if so how often did this appear in qualitative analysis?

**Limitations:**

The paper clearly discusses about the semi-realistic nature of the test environment and about the observed performance ceiling.

**Strengths And Weaknesses:**

The paper is well motivated by recent work on deceptive alignment, alignment faking, and constitutional classifiers (e.g., Hubinger et al., 2021; Carlsmith, 2023). It delves deep into the choice of optimization strategies for the task and clearly discusses the strengths and limitations. The choice to emphasize low-FPR monitoring through partial AUROC is also sensible for a real monitoring setting, where false positives quickly become operationally infeasible.

The main weakness is that the test setting still remains somewhat narrow relative to the paper’s broader framing around scheming. ControlArena is more realistic than the synthetic training pipelines, but the positive cases are still induced through injected secret side tasks in controlled sandbox environments (which makes its distribution very narrow and synthetic). That is useful, but it is not the same as naturally emerging scheming in deployment. The paper does acknowledge this as a semi-realistic setting. More broadly, it is still unclear how well the monitors would transfer to substantially different agent stacks, tools, or more subtle forms of long-horizon deception.

A second weakness is that the core method is more of a careful systems study than a strongly novel monitoring method. Stride and Gloom are useful pipelines, but the actual monitor is still a prompted frontier LLM over transcripts. The main innovation is therefore in data generation, evaluation design, and the empirical study of optimization behavior, rather than in a new monitoring architecture. This distinction is not clear from the title and abstract itself.

A third weakness is that the analysis of the performance ceiling remains suggestive rather than conclusive. The paper argues that overfitting arises because optimization starts exploiting synthetic artifacts, or because prompting may be fundamentally limited, but this remains largely an interpretation and not explored fully. A further exploration into disentangling whether the ceiling is mainly due to synthetic data quality, prompt-space limitations, or mismatch between optimization loss and the evaluation metric would have made the paper much stronger.

1. Hubinger, Evan, Chris Van Merwijk, Vladimir Mikulik, Joar Skalse, and Scott Garrabrant. "Risks from learned optimization in advanced machine learning systems." arXiv preprint arXiv:1906.01820 (2019).
2. Carlsmith, Joe. "Scheming AIs: Will AIs fake alignment during training in order to get power?." arXiv preprint arXiv:2311.08379 (2023).

---

> ### Author Rebuttal · Authors · 2026-03-31
>
> Thank you for your thoughtful and detailed review. We appreciate your recognition of the motivation of the work, the clarity of the optimization analysis, and the importance of evaluating monitoring performance in the low-FPR regime using partial AUROC. We also thank you for your constructive feedback. We address these points below and incorporate clarifications in the revised PDF.
>
> **Test set limitations.** We agree that our test set is limited by: (i) the finite set of sandbox environments (Table 2), and (ii) the fact that it is not fully realistic, as it relies on injected side tasks. Mitigating narrowness is fundamentally limited by the small number of scheming/sabotage environments currently in existence; however, we still include a diverse range of settings within this set, including SHADE Arena, itself a standalone benchmark. We agree that increased diversity would further improve evaluation. Regarding realism, since frontier models do not reliably exhibit egregious scheming in the wild, side-task injection is the most realistic available approach and reflects the current state of the art in control research. We reflect this level of realism by referring to the test set as “semi-realistic”.
>
> **Core contribution.** Our main contribution is an end-to-end framework: scalable synthetic data pipelines (STRIDE, Gloom), a suite of optimization approaches (with practitioner guidance), and a semi-realistic evaluation protocol with strict train-test separation, partial AUROC evaluation, and high-confidence execution-based labels.
> Why the performance ceiling was reached. We refer to our preliminary experiments and planned extensions described in our response to Reviewer 2gi4 (see "Diagnosing the performance ceiling."). We will include these results in the revised PDF.
>
> **DSPy vs. partial AUROC mismatch.** We agree that using example-wise classification loss for DSPy introduces objective mismatch with partial AUROC. An ablation is not feasible with GEPA or MIPROv2, which require example-wise losses. Empirically, DSPy optimization on strong monitors often leaves training pAUROC unchanged or decreased (Figures 16–17), indicating the mismatch indeed has some effect on optimization. Whilst human refinement relies on individual samples for generating candidate changes, prompts are assessed using partial AUROC directly as our target metric.  More broadly, the finding that performance plateaus beyond simple prompt sweeps is consistent across all optimization methods we tested, despite their differing objectives and procedures, which lends robustness to the conclusion. We have added clarifying remarks on these points in the revised PDF.
>
> **Insufficiency of action-only monitoring.** We compared action-only and Action+CoT monitors on ControlArena trajectories by evaluating a prompted monitor with and without CoT access. We found cases where action-only information (user messages, tool use, tool responses) were insufficient, while Action+CoT succeeded. The most common qualitative case was “blabbing,” where the agent revealed the side task in its thoughts; removing this information resulted in incorrect classification. We include this observation in the revised PDF. While CoT monitoring is an important direction, we believe focusing on action-only monitors remains well-motivated by the need for robustness to potential shifts in the availability or reliability of CoT.

---

> > ### Author Rebuttal · Reviewer_iDMb · 2026-04-02
> >
> > I will increase the score accordingly

---

### Official Review · Reviewer_bxU3 · 2026-03-17

**Soundness:** 2
**Presentation:** 3
**Significance:** 3
**Originality:** 2
**Overall Recommendation:** 3
**Confidence:** 3

**Summary:**

This work studies whether a black-box monitor can detect scheming from an agent’s observable trajectory alone, without access to its internal reasoning. They define scheming in natural language (which named as Constitutional), generate synthetic training data with two pipelines (STRIDE and Gloom), and use this data to optimize monitor prompts/models. They then evaluate on 7.5k held-out trajectories from ControlArena, a more grounded benchmark with 5 environments and deterministic labels based on whether a hidden malicious side task was actually completed. The main takeaway is that signals learned from synthetic data can transfer to a new realistic setting to some extent, but gains seem to plateau quickly.

**Compliance With Llm Reviewing Policy:**

Affirmed.

**Key Questions For Authors:**

1. I am unclear on the role of the “constitution” here. My reading is that it mainly serves as a natural-language specification used to generate synthetic data, while the monitor itself is still a prompted black-box classifier. In the cited prior work, the constitution seems to have a somewhat broader scope, shaping not just synthetic data generation but the overall safeguard setup more directly. Is that also the intended interpretation here? If so, it would help to clarify how much of the contribution is specifically constitutional.

**Limitations:**

yes

**Strengths And Weaknesses:**

# Strengths
1. The problem is important and well motivated that whether we can monitor an agent from its external behavior alone, rather than relying on CoT or internal activations.
2. The train/test split is thoughtfully designed.
A strong part of the paper is that the monitor is trained on synthetic data but evaluated on a separate benchmark, ControlArena, rather than being tested only on more data from the same generation pipeline. This makes the transfer claim more convincing.
3. The test labels are relatively clean.
I appreciated that the test set does not depend on LLM judges. Instead, positive examples are defined by whether the side task was actually completed in the environment. That makes the evaluation much easier to trust.

# Weaknesses

At a high level, the paper’s approach is straightforward: it defines a target class of misbehavior, uses frontier models to synthesize examples of that class, and evaluates whether a black-box monitor trained on this synthetic distribution transfers to a more grounded benchmark. The contribution is therefore primarily in the experimental pipeline and empirical study, rather than in a fundamentally new monitoring mechanism.

1. The paper uses a broad definition of scheming, but evaluates a much narrower version of it.
The paper defines scheming quite broadly, as deceptive pursuit of misaligned goals. But in the actual benchmark, it turns out to be a much narrower setup: the agent is given a side task, and the monitor is asked to detect trajectories where that side task is successfully carried out. That is a valid subclass of scheming-like behavior, but it is still much narrower than the general phenomenon suggested by the paper’s framing. In particular, it does not really address more subtle, self-initiated forms of deception.

2. The  **realism**  of the test setting should not be overstated. ControlArena is clearly more grounded than the synthetic training data,  but it is still created by injecting side tasks. So I do not think the results should be read as showing transfer to real deployment settings; they show transfer to a more realistic benchmark family.


3 Limited baselines. Most of the experimental comparison is within a single design family: prompted, action-only, single-LLM monitors. While this is a reasonable starting point, this work does not compare against other plausible black-box monitoring approaches, say, finetuned classifiers over trajectories,. Without such comparisons, it is difficult to know whether the reported ceiling is a ceiling of black-box monitoring itself, or only of the particular prompted-monitor setup studied here.

---

> ### Author Rebuttal · Authors · 2026-03-31
>
> Thank you for your thoughtful and detailed review. We appreciate your recognition of the importance of the problem, the strength of the train/test gap, and the use of clean, execution-based labels. We also thank you for the constructive feedback and questions, which we address below and incorporate into the revised PDF.
>
> **Core contribution.** We view our main contribution as an end-to-end framework for monitoring: pipelines for generating scalable synthetic training data (STRIDE and Gloom), a suite of optimization approaches (along with recommendations for practitioners), and a semi-realistic evaluation protocol with strict train-test separation, partial AUROC evaluation, and high-confidence execution-based labels.
>
> **Clarification of test setup.** While the test data is generated by assigning agents malicious side tasks, the monitors are not given access to the side task (or even told that one exists) during evaluation. Instead, monitors are provided with trajectories from the test set and are tasked with detecting scheming behaviour in general. The test set spans a diverse range of side tasks, all of which satisfy our general definition of scheming. Crucially, the side task instruction is not observable in the trajectory seen by the monitor, so the monitor cannot rely on task-specific cues and must instead detect general patterns of covert goal pursuit. The setting aims to emulate self-initiated scheming (i.e., the side tasks represent goals that may be instrumentally useful to a scheming agent, such as api key exfiltration). We hope this clarifies the concern around narrowness. We agree that the setup still does not capture all forms of real-world scheming and that a realism gap remains. We have added further clarification in the revised PDF and are happy to address any additional questions.
>
> **Test set realism.** We agree that the test setting is not fully realistic.  Since current frontier models do not reliably exhibit egregious scheming in the wild, side-task injection offers the most realistic approach currently available to us and reflects the current state of the art in control research. We intend not to overstate the realism and refer to our test set cautiously as semi-realistic.
>
> **Limited baselines.** While our focus was on prompt-based approaches, we agree that a fine-tuned baseline provides a useful comparison and can help disentangle whether the performance ceiling stems from the synthetic data or from the prompting paradigm itself. Since frontier model fine-tuning APIs block our dataset, we conduct this comparison entirely on open-source models. As an initial baseline, we fine-tuned Qwen3-8B (thinking enabled) on STRIDE and Gloom using GRPO with Brier rewards on the final score, using eg_v3 (Appendix G.1.1) as the prompt (batch size = 32, group size = 8, learning rate = 5e-6, LoRA rank = 32, 10 epochs). This yielded control arena (test set) partial AUROC scores of 0.549 (STRIDE trained) and 0.555 (Gloom trained). Fine-tuning substantially improved over the prompted Qwen3-8B base model (0.277), but did not surpass our best prompted frontier monitors. This already serves as a useful sanity check that simple fine-tuning of a smaller open-weight model does not trivially break through the observed performance ceiling. However, we do not consider this comparison complete. For the revised paper, we will extend the analysis in three ways. First, we will scale to a larger open-source model (e.g. Kimi K2 Thinking). Second, we will conduct hyperparameter tuning and report results across 3 runs with the best configuration to provide variance estimates. Third, we plan to apply prompt optimization to the same open-source model, giving us a like-for-like comparison of prompting versus fine-tuning on the same model class. Together, these experiments should provide substantially stronger evidence for whether the observed ceiling is a limitation of the synthetic data distribution or of the optimization approach. We have added our Qwen3-8B result in a new "Comparison to fine-tuning baseline" section, and will add the further fine-tuning experiments to the final PDF.
>
> **The role of the “constitution”.** In Constitutional Classifiers (Sharma et al., 2025), to our understanding the constitution is used only to generate synthetic training data. Our approach is analogous: both STRIDE and Gloom use a constitution specifying the nature of scheming to produce labeled trajectories, and the resulting monitors are optimized on this data but do not directly access the constitution during optimization or at inference time. The constitution also incidentally guides prompt design and test-set validation (we ensure side tasks meet our scheming definition), but these are secondary to its primary function as a data generation specification. We will clarify this in the revised PDF.
>
> We appreciate your feedback and believe these clarifications will strengthen the paper. We would be happy to address any further questions.

---

> > ### Author Rebuttal · Reviewer_bxU3 · 2026-04-04
> >
> > Thank authors for the rebuttal.
> >
> > Regarding the role about `constitution`,  I still think the analogy to (Sharma et al) is somewhat overstated. In (Sharma et al), the constitution feels more self-contained and central to the safeguard design: it serves as an explicit policy specification that defines the classifier’s target behavior and supports adaptation to new threat models, rather than merely acting as a recipe for data generation. By contrast, in this paper the constitution appears to play a narrower role, mainly specifying the scheming concept used to synthesize training examples, with only secondary influence on prompt design and validation. So while the comparison is directionally reasonable, I do not think the constitutional aspect is equally substantive in the two works.
> >
> > I do not view this as a fatal flaw, but I do think it affects how the contribution should be positioned. For this reason, I am happy to change my score to "borderline" if there is one, and am not changing my score right now. If this paper ends up near the acceptance boundary, I would welcome further discussion if the AC thinks this framing issue is decision-relevant.

---

> > > ### Author Response · Authors · 2026-04-05
> > >
> > > Thank you for the thoughtful follow-up, we appreciate you engaging with this point in detail.
> > >
> > > We think there may still be some mismatch in how we are interpreting the role of the “constitution,” so we’d like to clarify our intended claim and address what we take to be the most likely readings of your concern.
> > >
> > > Our intended use of “constitutional” is a narrow one, aligned with the description in Constitutional Classifiers: “safeguards trained on synthetic data, generated by prompting LLMs with natural language rules (i.e., a constitution) specifying permitted and restricted content.” In our setup, the constitution likewise specifies scheming vs. non-scheming behavior and is used to generate labeled trajectories, on which the monitor is optimized.
> > >
> > > Concretely, our constitution defines scheming behavior and we generate both:
> > > - **positive examples** that satisfy this definition, and
> > > - **negative examples** consisting of aligned/non-scheming trajectories that do not satisfy it.
> > >
> > > In this sense, the constitution induces a full decision boundary between permitted and restricted behavior, analogous to the harmful vs. harmless delineation in Sharma et al., though expressed through a single behavioral definition rather than multiple explicit categories.
> > >
> > > We agree that this may not have been sufficiently clear in our framing, and we will clarify this more explicitly in the revised version.
> > >
> > > **On the role of the constitution.**
> > > You mention that in Sharma et al. the constitution “feels more self-contained and central to the safeguard design” and that it “serves as an explicit policy specification … rather than merely acting as a recipe for data generation,” and conclude that “the constitutional aspect is [not] equally substantive in the two works.”
> > >
> > > From our reading, in both works the constitution:
> > >
> > > - defines the target behavior (permitted vs. restricted), and
> > > - is operationalized through synthetic data generation used to optimize a classifier/monitor, rather than being directly consulted at inference time.
> > >
> > > One possible difference is that Sharma et al. presents the constitution as multiple explicit categories, whereas we use a single high-level definition of scheming. This also means our definition is more abstract, while theirs is more concretely decomposed into specific behavioral categories. However, this is a difference in granularity and presentation rather than mechanism; our definition could be decomposed into multiple subcategories without changing how the system is optimized or used.
> > >
> > > Regarding the points you raise about supporting adaptation to new threat models and defining the classifier’s target behavior, in both works this is achieved through modifying the constitution and generating more synthetic data. In our setup, this corresponds to refining the scheming definition or adding new categories and generating additional data accordingly. As in Sharma et al., these capabilities arise through the data generation process rather than a separate mechanism.
> > >
> > > **On positioning.**
> > > Finally, our intention was not to claim equal substantiveness of the constitutional aspect, but to situate our approach within prior work that uses natural-language specifications to generate synthetic training data for safeguards.
> > >
> > > In summary, we maintain that the role of the constitution and the mechanism by which it is operationalized are consistent between the two works. We acknowledge differences in presentation and level of granularity between our constitution and that in Sharma et al., and will add clarifying comments accordingly in the revised PDF.

---

### Decision · Program_Chairs · 2026-04-30

**Decision:**

Accept (regular)

**Comment:**

The paper studies detecting scheming behavior in LLM agents. The authors propose constitutional black-box monitors to detect scheming behavior in LLM agents. They also introduce two synthetic data generation pipelines STRIDE and Gloom to generate synthetic trajectories for such monitors. Authors show that monitors optimized on synthetic data can generalize to realistic settings but performance saturates quickly.

Reviewers agreed that the problem is important and well-motivated. Having two methods to generate synthetic data was also found to be helpful to strengthen the findings of the paper. Reviews also appreciated the experimental design choices of the paper. Reviewers also found findings of the paper insightful and meaningful. However, there were some concerns about limited novelty of the work along with concerns around the way constitution was framed in this work which makes the paper overstate the analogy to constitutional classifiers. Though the reviewer acknowledges this as "not a fatal flaw".